# SAMoE-VAE: A Tabular Foundation Model with a Schema-Aware Mixture-of-Experts Variational Autoencoder

## ABSTRACT

Foundation models have revolutionized vision and language, yet tabular learning still depends on bespoke, per-dataset pipelines. A key challenge in developing a uniform representation that enables foundation model is *schema mismatch*: real-world tables contain diverse column types: numeric, categorical, text, datetime, whose semantics vary across datasets. We frame cross-tabular representation learning as a weakly supervised, multi-modal problem, leveraging the readily available schema metadata that accompanies each table. We propose SAMoE-VAE, a schema-aware Mixture-of-Experts VAE that: (i) assigns separate experts to numeric, categorical, text, and datetime columns; (ii) fuses expert posteriors via a schema-conditioned Product-of-Experts(MoPoE); (iii) produces a probabilistic latent embedding space that drives accurate downstream prediction and schema-aware generation. To train at scale, we curate **Meta-T4**, a 1.2-million-table corpus augmented with LLM-generated text metadata. Extensive experiments show that SAMoE-VAE outperforms prior art in tabular foundation models on representation learning benchmarks, yielding higher downstream accuracy and improved sample efficiency.

## 1 INTRODUCTION

Tabular data forms the basis of decision-making in virtually every industry, empowering applications from clinical trial cohort synthesis and patient risk stratification to financial portfolio optimization and supply-chain forecasting (Rajkomar et al., 2018; Choi et al., 2016; Fischer & Krauss, 2018). Foundation models in vision and NLP have shown that a single, large pretrained network can be fine-tuned or prompted to solve myriad downstream tasks with minimal additional effort (Brown et al., 2020; Dosovitskiy et al., 2021; Devlin et al., 2019; Kolesnikov et al., 2020; Radford et al., 2021; Ramesh et al., 2021). Analogously, a tabular foundation model would enable shared representations across datasets, drastically reducing per-dataset engineering overhead, accelerating transfer learning, and supporting zero-shot prediction and data synthesis in resource-constrained domains (Wang & Sun, 2022; Yoon et al., 2020; Kim et al., 2024; Hollmann et al., 2025; van Breugel & van der Schaar, 2024)

However, developing such foundation models faces a significant challenge: the absence of a canonical, instance-level representation across datasets due to heterogenous schemas. In language or vision, examples can be mapped into a uniform input space (tokens or pixel grids) with task-agnostic pre-processing (Grinsztajn et al., 2022; Battaglia et al., 2018); in tables, the schema itself encodes types, domains, distribution pattern, etc, that govern valid operations(Yin et al., 2021; Deng et al., 2022). Studies such as TAPAS (Herzig et al., 2020), TURL (Deng et al., 2022), and the relational-inductive-bias framework (Battaglia et al., 2018) argue that learning algorithms must respect this schema information to transfer across tables and reason compositionally."For tabular foundation models, database schemas provide machine-readable, low-cost supervision (types, domains, ordinality, units, keys) that is rarely available in standardized form in vision/text." (van Breugel & van der Schaar, 2024). E.g., a patients table might declare columns age: int ($\geq 0$), blood_type: enumA, B, AB, O, and hospital_id: foreign_key—each with distinct type, domain, and provide rich context supervision. Yet many current tabular foundation models linearize cells into discrete tokens (Hegselmann et al., 2023; Wang et al., 2023; Hollmann et al., 2025) or aggregate per-column

embeddings (Kim et al., 2024; Yang et al., 2024; Lin et al.), often without representing column types , ordinality, table context and so on. This limits the transfer of learned representation to unseen schemas.

We frame cross-table representation learning as a metadata-supervised, multi-modal problem: databases routinely expose machine-readable schema metadata (column types, semantic meaning, and distribution patterns) that can guide specialization across heterogeneous tables where modalities (numeric, categorical, temporal, text) and even their presence/absence vary by dataset. We introduce **Schema-Aware MoE-VAE (SAMoE-VAE)**, a variational auto-encoder in which a low-dimensional learned *schema vector* gates specialized encoder–decoder experts. By incorporating column types, domain tags, distribution patterns, and other metadata directly into the routing mechanism, SAMoE-VAE (i) preserves modality distinctions during encoding, (ii) synthesizes realistic tables for previously unseen schemas, and (iii) provides expert-specific latent priors that capture epistemic uncertainty while keeping inference cost low. Unlike prior work that encoded column information implicitly as column name embedding (Wang & Sun, 2022; Kim et al., 2024; Lin et al.) or textual description (Wang et al., 2023), our method addresses the multi-modality and heterogeneity at deeper architectural level by routing different data types to experts and learns their weighting dynamically via mixture-of-product-of-expert mechanism. Embedding the Mixture-of-Experts directly within the VAE's encoder–decoder yields a smooth, continuous posterior leveraging the theoretical guarantees of variational inference for coherent probabilistic representations (Rezende et al., 2014).

We present three key contributions:

1. **Schema–Aware MoE–VAE.** We propose the first variational auto–encoder that employs a schema–gated Mixture–of–Experts layer for tables, preserving modality distinctions during encoding and enabling schema–conditioned synthesis.

2. **Meta–T4 Metadata Benchmark.** We extend the existing T4 corpus by automatically generating, via LLMs, fine-grained table- and column-level context descriptions and column-type annotations—creating the first large-scale tabular dataset of its kind enriched with comprehensive schema metadata.

3. **Schema-generalization learning formulation.** We cast tabular foundation modeling as a metadata-supervised, multi-modal problem and show that schema metadata is a strong supervisory signal, improving representation quality, generation fidelity, and zero/low-shot transfer to unseen schemas

Together, these components move toward web-scale, schema-robust tabular models and set the stage for latent-diffusion foundation models capable of high-fidelity conditional generation under unseen schemas

## 2 RELATED WORK

### 2.1 SELF-SUPERVISED LEARNING ON TABULAR DATA

Recent efforts toward tabular "foundation" modeling can be usefully grouped into three families based on their input representation:

**(1) Table LLMs (flatten-to-text).** These methods serialize rows (or row–column tuples) into natural-language sequences and leverage general-purpose LLMs via masked-cell or instruction-style objectives, with schema cues provided through column names or brief descriptions. TabLLM (Hegselmann et al., 2023) and UniPredict (Wang et al., 2023) convert each row or row–column triplet into sentences and fine-tune LLMs accordingly; TabT5 (Narayan et al., 2024) adopts a T5 backbone with schema-aware prompting; Text2Table (Li et al., 2024) probes numeric reasoning by generating QA pairs over serialized rows; TabRewrite (Garcia et al., 2025) injects external knowledge through retrieval-augmented rewriting; and Tabula-8B (Gardner et al., 2024) adapts a pre-trained LLM with a large table corpus and block attention, yielding strong zero-/few-shot classification. While this paradigm benefits from broad NLP transfer, flattening continuous values into subword tokens and foregoing modality-specific processing can harm fidelity and calibration on mixed-type tables.

**(2) Token-wise in-context learners.** Here, compact Transformers are trained on large suites of synthetic tasks—often sampled from Bayesian networks—to learn in-context adaptation over tokenized examples, optionally augmented with schema tokens from column names or prompts. TabPFN and its scalable variants (Hollmann et al., 2025; Ye et al., 2025; Feuer et al., 2024; Helli et al., 2025) fit prior data distributions and infer feature relations on the fly; TabPrompt (Zhang et al., 2024b) conditions prompts on column names to strengthen transfer; and Schema2Vec (Dimitriadis et al., 2025) encodes column metadata as additional tokens. The approach often excels in small-data supervised regimes without per-dataset training, but it does not natively encode explicit type or constraint signals, lacks architectural support for multimodal columns (e.g., text, images, audio), and remains discriminative rather than generative.

**(3) Latent-embedding models.** These models operate on native mixed types through modality-aware encoders and learn amortized representations via reconstruction or self-supervision, typically enriching schema signals with embeddings of names, types, or lightweight tags. Representation-centric approaches like CARTE (Kim et al., 2024) and CTSyn (Lin et al.) build a shared latent space with reconstruction losses: CARTE employs a graph-attentional star-graph over rows, TransTab encodes text embeddings of feature values and names with a single Transformer, and CTSyn fuses schema embeddings into a conditional latent-diffusion autoencoder. UniTabE (Yang et al., 2024) organizes table elements into "TabUnit" modules for masked-value prediction to capture cross-schema patterns; TP-BERTa (Yan et al., 2024) tokenizes numeric magnitudes with intra-feature attention and rivals GBDTs; and CV2 (Ye et al., 2024) aligns cell embeddings with column metadata via a contrastive masked-context objective. Despite these advances, most methods still treat cells uniformly beyond column-name/type cues and lack explicit *routing* of heterogeneous modalities to specialized experts or schema-gated pathways.

**Our departure.** SAMoE-VAE assigns dedicated experts to distinct data modalities and employs a learned *schema vector* to *gate* them. This design integrates rich metadata—types, missingness patterns, domain tags, and inter-column signals—into a probabilistic latent space (via MoPoE) that supports high-fidelity generation, principled uncertainty, and schema-conditioned transfer, without brittle tokenization or fixed-schema assumptions.

**MoE in tabular models.** Mixture-of-Experts (MoE) dates to adaptive gating (Jacobs et al., 1991) and sparse Transformers (Shazeer et al., 2017; Fedus et al., 2021); multimodal VAEs often dispatch *fixed* modalities (image/text/audio) to experts. In tabular settings, **TabMoE** (Wu & Hou, 2024) routes queries to *task*-specific experts for table QA; **GG-MoE** (Chernov, 2025) uses Gumbel-Softmax over column embeddings for classification with per-dataset gates; and **Tabby** (Cromp et al., 2024) assigns experts per *column* for data synthesis. These lines do not treat numeric/categorical/text/datetime as first-class *modalities* nor tie routing to machine-readable *schema* that varies across datasets. By contrast, **SAMoE-VAE** (i) assigns experts by *column type* and (ii) computes mixture weights from a learned *schema vector*, enabling schema-conditioned specialization; further, it integrates MoE within a *generative* VAE using a MoPoE latent aggregator, supporting uncertainty-aware representation and schema-aware generation.

## 3 METHODOLOGY

In this section, we detail the design of SAMoE-VAE workflow, as shown in figure 1.

### 3.1 FEATURE EMBEDDING

Given a row $\boldsymbol{x} = (c_1, x_1, \ldots, c_p, x_p)$ with $p$ columns, we embed names and values into a shared $M_{\text{LM}}$-dimensional space. For each column $i$: $e_{c_i} = \text{LM}(c_i)$; and

$$
e_{x_i} = \begin{cases}
\text{LM}(x_i) & \text{(categorical)} \\
W_{\text{num}}\text{PLE}(x_i) & \text{(numeric)} \\
W_{\text{text}}\text{AE}_{\text{text}}(x_i) & \text{(free text)} \\
W_{\text{time}}\text{CycEnc}(x_i) & \text{(datetime)}
\end{cases}
$$

where LM is a frozen text encoder. Note that categories are tokenized and encoded with LM. Numerical values are encoded using piece-wise linear encoding (PLE) (Gorishniy et al., 2022)

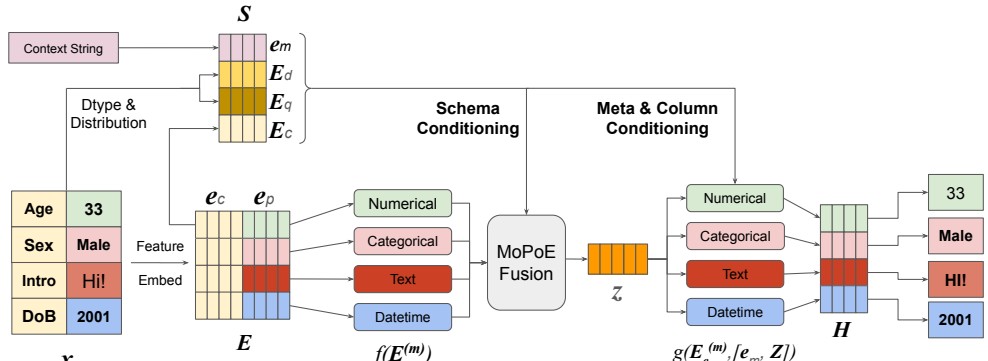

Figure 1: Overview of the SAMoE-VAE workflow. A table row, its column names and metadata are first embedded into a unified sequence $E$. This sequence is split by data type (numeric, categorical, text, datetime) and processed by corresponding expert encoders to produce modality-specific Gaussian parameters. In parallel, schema metadata (table description, column names, data types, value distributions) is encoded into a schema vector $s$. A schema-conditioned product-of-experts gate fuses expert outputs into a single latent $z$. Finally, $z$, the metadata, and column names are routed to modality-specific decoders to reconstruct each cell's value.

and linear projection to $M_{\text{LM}}$ is learned. Text values are embedded using a BART (Lewis et al., 2020)-based autoencoder (Lovelace et al., 2024) whose encoder output is compressed via a Perceiver-Resampler (Alayrac et al., 2022) into $\mathbb{R}^{M_{\text{LM}}}$. Datetime values are first decomposed into year, month, day, hour, and minute components; these are encoded using sine–cosine positional encodings and mapped to $M_{\text{LM}}$ via a two-layer MLP. (Suh et al., 2024). We then follow a framework similar to CTSyn (Lin et al.) to concatenate each column name embedding $e_{c_i}$ with its corresponding value embedding $e_{x_i}$, forming the final sequence:

$$\boldsymbol{E} = \big[\, [e_{c_1}; e_{x_1}],\, [e_{c_2}; e_{x_2}],\, \ldots,\, [e_{c_p}; e_{x_p}]\,\big] \in \mathbb{R}^{p \times 2M_{\text{LM}}}.$$

Perceiver cross-attention naturally handles variable-length $\boldsymbol{E}^{(m)}$ without padding to a common length; batching uses standard key/query masks.

## 3.2 Modality–Specific Expert Encoders

We first split the embedding matrix $\boldsymbol{E} \in \mathbb{R}^{p \times 2M_{\text{LM}}}$ by column data type into sub-sequences $\{\boldsymbol{E}^{(m)}\}_{m \in \mathcal{M}}$, where $\mathcal{M} = \{\text{num}, \text{cat}, \text{text}, \text{time}\}$. This ensures that all modality–specific inputs share the same embedding width ($2M_{\text{LM}}$) while grouping values by their true semantic type.

Each $\boldsymbol{E}^{(m)}$ is then passed to its own Perceiver–Resampler expert $f_m$ (identical architecture), which uses learnable latent queries and cross-attention to transform a variable-length sequence $\boldsymbol{E}^{(m)}$ into a fixed-size tensor

$$(\mu^{(m)}, \log \sigma^{2(m)}) = f_m(\boldsymbol{E}^{(m)}), \qquad \mu^{(m)}, \log \sigma^{2(m)} \in \mathbb{R}^{\ell \times d_z}.$$

where $\ell$ is the number of latent queries per modality and $d_z$ is the latent dimensionality. The resulting $\mu^{(m)}$ and $\log \sigma^{2(m)}$ parameterise modality-specific diagonal-Gaussian posteriors, which are fed directly into the schema-aware MoPoE gate (Sec. 3.3). No latent sampling occurs at this stage, and permutation invariance is preserved by omitting positional embeddings. Missing modalities are skipped.

## 3.3 Schema–Aware Mixture–of–Product–of–Experts Gating

**Intuition.** Product-of-Experts (PoE) yields a tighter, modality-consistent posterior than simple averaging and is closed-form for Gaussians (Wu & Goodman, 2018). Unlike classic multimodal VAEs with a fixed modality set (e.g., image+text), tabular data vary in both the *presence* and *count* of

each modality. We therefore compute one Gaussian posterior per subset of available modalities and *gate only at fusion*: a schema-conditioned network (driven by the schema vector) assigns mixture weights over these subsetwise PoE posteriors. Figure 2 illustrates the schema-aware fusion.

**Schema vector $s$: construction.** We build three token streams (matrices in $\mathbb{R}^{p \times M_{\text{LM}}}$) from the row-level inputs:

$$\boldsymbol{E}_c = [e_{c_1}, \ldots, e_{c_p}], \quad \boldsymbol{E}_d = [e_{d_1}, \ldots, e_{d_p}], \quad \boldsymbol{E}_q = [e_{q_1}, \ldots, e_{q_p}].$$

**Name tokens $\boldsymbol{E}_c$:** $e_{c_i} = \text{LM}(c_i)$ (frozen text encoder, cached). **Type-ID tokens $\boldsymbol{E}_d$:** one-hot column types {num, cat, text, time} embedded via a learned lookup $E_{\text{type}} \in \mathbb{R}^{4 \times M_{\text{LM}}}$, so $e_{d_i} = E_{\text{type}}[\text{type}(i)]$.

**Distribution tokens $\boldsymbol{E}_q$** summarize per-column value distributions by modality: for *numeric* columns, we compute a 33-bin histogram over standardized values (z-scores), apply the **Discrete Cosine Transform (DCT-II)** to the normalized histogram, retain the first $K$ low-frequency coefficients that capture overall shape, and linearly project to $M_{\text{LM}}$ to obtain $e_{q_i}$; this yields a compact, shift/scale-insensitive signature that reflects properties like unimodality vs. multimodality and skew or heavy tails without depending on column length. For *categorical* columns, we take the frequency-weighted mean of category embeddings from LM and project to $M_{\text{LM}}$. For *datetime* columns, we build cyclic histograms (e.g., hour-of-day, day-of-week), form sine–cosine features, and project to $M_{\text{LM}}$. For *text* columns, we use the mean of $\text{AE}_{\text{text}}$ embeddings over observed cells and project to $M_{\text{LM}}$.

We concatenate $[\boldsymbol{E}_c; \boldsymbol{E}_d; \boldsymbol{E}_q]$ (no positional encodings) and pass through a lightweight Perceiver (4 layers, 2 heads) to produce the schema vector $\boldsymbol{s} \in \mathbb{R}^{M_{\text{sch}}}$ (we use $M_{\text{sch}}{=}256$).

**Element-wise Product of Experts.** Let $\mathcal{M}$ be the modality index set and $\mathcal{P}(\mathcal{M}) \setminus \{\emptyset\}$ its non-empty subsets. For each subset $S$ we compute a diagonal-Gaussian product posterior

$$q_S(\boldsymbol{z} \mid \boldsymbol{x}_S) = \mathcal{N}\big(\mu_S, \text{diag}(\sigma_S^2)\big), \qquad \sigma_S^{-2} = \sum_{m \in S} \sigma_m^{-2}, \ \mu_S = \sigma_S^2 \sum_{m \in S} \sigma_m^{-2} \mu^{(m)},$$

applied *element-wise* across the $\ell \times M_{\text{agg}}$ latent matrix. With four modalities we have 15 subsets, a tractable number. If a modality is absent, its expert outputs the unit prior and is down-weighted.

**Schema-conditioned gating.** Each product mean $\mu_S$ is mean-pooled over its $\ell$ tokens and concatenated with $\boldsymbol{s}$; a two-layer MLP ($M_{\text{agg}}{+}M_{\text{sch}} \to 128 \to 1$, ReLU) produces logits $a_S$. Softmax gives mixture weights $\alpha_S = \text{softmax}(a_S)$; omitting $\boldsymbol{s}$ collapses the gate to vanilla MoPoE.

**Mixture aggregation.** The final posterior is an element-wise precision-weighted Gaussian

$$\sigma_*^{-2} = \sum_S \alpha_S \, \sigma_S^{-2}, \qquad \mu_* = \sigma_*^2 \sum_S \alpha_S \, \sigma_S^{-2} \mu_S,$$

from which we draw $\boldsymbol{z} = \mu_* + \sigma_* \odot \varepsilon, \ \varepsilon \sim \mathcal{N}(0, I)$ follow reparameterization trick.

## 3.4 DECODER

**Latent-to-cell decoding.** For each modality $m \in \{\text{num}, \text{cat}, \text{text}, \text{time}\}$, we combine the shared row latent $\boldsymbol{z} \in \mathbb{R}^{\ell \times M_{\text{agg}}}$ with the table-level metadata $e_m$ and the modality-specific column-name embeddings $\boldsymbol{E}_c^{(m)} = [e_{c_i}]_{i \in m}$. A modality-specific Perceiver–Resampler $g_m$ (identical architecture across modalities but not weight-tied) maps queries $\boldsymbol{E}_c^{(m)}$ and keys/values $[e_m; \boldsymbol{z}]$ to cell embeddings:

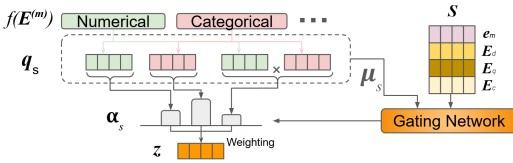

Figure 2: Schema-aware Mixture-of-product-of-expert gating.

$$\boldsymbol{H}^{(m)} = g_m\big(\boldsymbol{E}_c^{(m)}, [e_m; \boldsymbol{z}]\big) \in \mathbb{R}^{p_m \times M_{\text{dec}}},$$

where $p_m$ is the number of columns of type $m$. Each cell vector $h_i^{(m)}$ is then passed to a lightweight, modality-specific reconstruction head: a linear layer with sigmoid activation for numerics, cosine-similarity softmax over prototype embeddings for categoricals, a Perceiver–Resampler + BART

decoder for text, and multi-task MLPs for datetime (year via MSE, others via cross-entropy). We compute the total reconstruction loss as

$$\mathcal{L}_{\text{rec}} = \sum_m \sum_{i \in m} \ell^{(m)}\big(\hat{x}_i^{(m)}, x_i\big).$$

Using separate decoders per modality aligns with the Mixture-of-Experts principle by allowing each expert to specialise in reconstructing its own input distribution, reducing interference across heterogeneous column types.

## 3.5 TRAINING OBJECTIVE

The VAE minimises

$$\mathcal{L} = \underbrace{\sum_m \sum_{i \in m} \ell^{(m)}(\hat{x}_i, x_i)}_{\mathcal{L}_{\text{rec}}} + \beta(t) \underbrace{\sum_{k=1}^{\ell M_{\text{agg}}} D_{\text{KL}}\big(\mathcal{N}(\mu_k, \sigma_k^2) \,\|\, \mathcal{N}(0,1)\big)}_{\mathcal{L}_{\text{KL}}} + \lambda_{\text{con}} \underbrace{\mathcal{L}_{\text{con}}}_{\substack{\text{Info-NCE over} \\ \text{row latents}}},$$

where $\beta(t)$ is cosine annealed from 0 to 0.0001 over the total number of steps—balancing regularisation of the $\ell \times M_{\text{agg}}$ latent space with reconstruction fidelity, and $\lambda_{\text{con}} = 0.001$ was chosen via grid search over $\{0.1, 0.01, 0.001\}$. The supervised contrastive loss (Oord et al., 2018) is applied at every step, grouping row latents by class label (or quartile for regression) to enhance downstream separability. To promote robustness, we randomly mask 15 % of all cell embeddings (uniformly across modalities and respecting same-table batching) before decoding, while still computing $\mathcal{L}_{\text{rec}}$ on the full table.

**Large-Scale Pre-training.** We pre-train SAMoE-VAE on the Meta-T4 dataset, which consists of T4 corpus (Gardner et al., 2024) (1.2 M tables, 3.1 B cells) and LLM-annotated metadata. For each table, we present the first 30 rows in string form to the Llama-3 8B model (Grattafiori et al., 2024) to generate structured metadata including a table-level description, column-wise semantics, and type annotations; all textual fields are embedded with GTE-large (Li et al., 2023). Both modality-specific encoders and the decoder use six cross-attention layers with latent width $M_{\text{agg}} = 128$ and latent length $\ell = 64$. We optimise with AdamW (learning rate $1 \times 10^{-4}$, weight decay $1 \times 10^{-4}$) and cosine annealing (5% warm-up, floor $5 \times 10^{-5}$). Mini-batches contain 128 rows drawn from the same table, with 4-step gradient accumulation yielding an effective batch size of 512. Training is performed on a slurm cluster with single H100 GPUs for 240 hours; to maximize coverage of diverse schema, we randomly sample 256 examples from each table, resulting in a training subset covering about 10% of row instances but 100% schema in T4 dataset. Additional hyperparameters and hardware details are provided in Appendix 6.3.

## 4 EXPERIMENT

In this section we perform extensive experiments on the schema-aware MoPoE VAE method in table representation and synthetic table generation tasks. We seek to answer the following questions: (1) Does our method learn strong cross-table representations? (2) Is schema-aware MoPoE critical to that performance?

## 4.1 BENCHMARK DATASETS

We evaluate on three established suites that stress *schema–generalization* across heterogeneous mixed–type tables. First, following (Gardner et al., 2024), we use **OpenML–CTR23** (Fischer et al., 2023) as a *regression* suite that retains the original continuous targets; official train/test splits are used throughout. Second, **OpenML–CC18** provides a diverse set of mixed–type *classification* datasets with the standard OpenML partitions. Third, the **UniPredict** collection (Wang et al., 2023) offers additional *classification* tables curated for LLM–based tabular evaluation; we follow the authors' splits to enable comparison with "flatten–to–text" approaches. To probe sample efficiency, for each dataset we subsample the training portion at sizes $\{8, 16, 32, 64, 128\}$ plus the full set, repeating each size with three random seeds and reporting seed–averaged metrics. Classification is reported as accuracy and AUROC (macro when applicable); regression as RMSE.

## 4.2 BASELINES

We compare **SAMoE–VAE** against representative methods from the three families in Sec. 3.1, plus a strong conventional encoder, and we state for each whether we evaluate *embeddings* via a linear/logistic regression and/or *end–to–end* predictions. **CARTE** (Kim et al., 2024) (graph–attentional cross–column encoder), **CTSyn** (Lin et al.) (VAE backbone with schema embeddings), **Transformer VAE** (Zhang et al., 2024a) (modality–specific encoders for mixed types), and **SwitchTab** (Wu et al., 2024) (VAE with data–specific structural experts) are used as *latent–embedding* foundations and evaluated with *embeddings+regression*. For token–wise in–context learners we use **TabPFN v2** (Hollmann et al., 2025) and report its *end–to–end* predictions; when feasible we also extract intermediate representations and apply the same probe protocol for a fair comparison. For table LMs, we include **Tabula–8B** (Gardner et al., 2024) and **TAPAS** (Herzig et al., 2020), reporting *embeddings+regression* performance. As a conventional deep encoder we use **TabVec** (Skrub, 2024) over one–hot/normalized features with *embeddings+probe*. In all embedding settings, we freeze the backbone after trained on train set of each table in the benchmarks, fit a linear probe (logistic for classification; ridge for regression) on training embeddings, and evaluate on the test split; for TabPFN v2 and Tabula–8B we additionally report their native end–to–end predictions. Our goal is not to supersede specialized discriminative SOTA (e.g., TabPFN/XGBoost), but to assess whether *schema–aware* MoPoE yields stronger and more transferable *representations* across schemas.

## 4.3 DOWNSTREAM PERFORMANCE

Table 1: Benchmarking Results Across Tabular Representation Models (with Std. Dev. of Avg. Rank). Avg. rank computed over embedding models only; raw-feature end-to-end rows are shown for context.

| **Model** | CTR-23 | CC18 | | UniPredict | | **Avg. Rank** |
|---|---|---|---|---|---|---|
| | RMSE (↓) | ROC AUC (↑) | F1 (↑) | ROC AUC (↑) | F1 (↑) | (Std.) |
| TabVec | 1.532 | 0.681 | 0.412 | 0.472 | 0.321 | 5.70 (2.17) |
| TaPas | 3.658 | 0.592 | 0.298 | 0.593 | 0.305 | 10.00 (0.71) |
| Tabula-8B | 5.757 | 0.608 | 0.226 | 0.601 | 0.321 | 7.50 (1.32) |
| TabSyn-VAE | 1.145 | 0.821 | 0.307 | 0.607 | 0.289 | 5.20 (2.17) |
| SwitchTab | 1.039 | 0.914 | 0.423 | 0.699 | 0.408 | 2.00 (0.71) |
| CTSyn | 1.542 | 0.658 | 0.361 | 0.643 | 0.347 | 5.40 (1.14) |
| CARTE | 1.758 | 0.783 | 0.152 | 0.676 | 0.142 | 6.80 (2.28) |
| TabPFN | 1.429 | 0.808 | 0.396 | 0.702 | 0.381 | 3.40 (0.89) |
| TabPFN (raw features) | **0.961** | **0.942** | **0.820** | 0.859 | 0.718 | – |
| XGBoost (raw features) | 0.973 | 0.925 | 0.795 | 0.843 | **0.739** | – |
| CARTE (raw features) | 1.124 | 0.891 | 0.742 | 0.684 | 0.546 | – |
| SAMoE-VAE | 0.991 | 0.859 | 0.451 | **0.892** | 0.677 | 1.20 (0.45) |

We evaluate each embedding along one primary dimension. For each dataset, we compute embeddings of the predictor features and fit a logistic/linear regression on the training embeddings, then report RMSE for regression (CTR-23) and AUROC/macro-F1 for classification (CC18/UniPredict). By using the same regression model, we can isolate the embeddings' representational power; exact hyperparameters are detailed in Appendix 6.4.

From Table 1, SAMoE-VAE is the top performer among embedding baselines across benchmarks, achieving the best overall average rank. Strong prediction pipelines using raw features (e.g., TabPFN and XGBoost) still set the state of the art in downstream classification, which aligns with their design goal; our objective is different: produce the best general-purpose tabular embeddings. On the text-heavy **UniPredict** suite, SAMoE–VAE attains the strongest embedding performance, consistent with its dedicated text/datetime encoders; see App. 6.9 for a focused case study.

## 4.4 CROSS-SCHEMA GENERALIZATION

### 4.4.1 TABLE RECONSTRUCTION

We evaluate the ability of SAMoE-VAE as a foundation model to generalize to unseen table schemas. Table 2 compares SAMoE-VAE with strong VAE baselines on reconstruction of a representative

Table 2: Reconstruction Quality Across VAE Models

| Variant | Column Shape (↑) | Column Corr. (↑) | NRMSE (↓) | Cat. Acc. (↑) |
|---|---|---|---|---|
| SAMoE-VAE (fine-tuned) | 0.96 | 0.89 | 0.011 | 0.996 |
| SAMoE-VAE (zero-shot) | 0.90 | 0.79 | 0.019 | 0.834 |
| TabSyn-VAE | 0.93 | 0.76 | 0.017 | 0.995 |
| SwitchTab AE | 0.92 | 0.71 | 0.018 | 0.974 |

CTR-23 dataset with their own decoders (additional results in Appendix 6.4). *Column Shape* is $1-$KS, the complement of the Kolmogorov–Smirnov distance (mean over numerical columns); *Column Corr.* is the mean absolute Pearson correlation between reconstructed and true numerical columns (both metrics therefore lie in $[0, 1]$, with higher better). NRMSE denotes per-column normalized RMSE; classification is reported as *Cat. Acc.* on categoricals. The main downstream table (Table 1) reports dataset-level RMSE for CTR-23. Without any gradient updates ("zero-shot"), the pretrained SAMoE-VAE already achieves strong shape and correlation metrics, indicating that its latent captures realistic column distributions even for unseen schemas. Fine-tuning on each training set further improves all reconstruction metrics—surpassing both TabSyn-VAE and SwitchTab-AE—showing that schema-aware gating yields representations that transfer across tables yet refine quickly when modest in-domain data are available. This robust zero-shot reconstruction ability lays the foundation for integrating our model into broader generative frameworks, such as latent-diffusion pipelines.

### 4.4.2 DATA EFFICIENCY

Figure 3 shows classification ROC AUC as we vary the number of labeled training examples from 8 to 128. SAMoE-VAE (orange) delivers the strongest performance in the low-data regime—outperforming all baselines by a wide margin at 8 and 16 examples and maintaining a lead through 32 examples. As the training set grows, TabPFN (red) and SwitchTab (green) recover rapidly and slightly overtake SAMoE-VAE at 64 and 128 samples, while CARTE and TAPAS remain at the bottom of the curve. This behavior reinforces that the schema-aware mixture-of-experts embedding learned by SAMoE-VAE is markedly more sample-efficient, extracting useful features when labels are scarce, yet remains competitive as data scales.

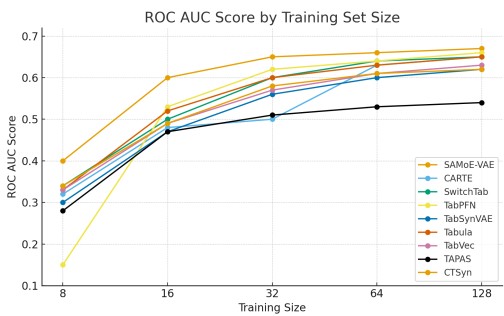

Figure 3: Classification AUROC across training set sizes. Each curve represents a different tabular representation model.

### 4.5 ROBUSTNESS TO METADATA QUALITY

Table 3: UniPredict ROC AUC for models with/without textual metadata

| Metadata | SAMoE-VAE | Tabula-8B | CARTE |
|---|---|---|---|
| Without Meta | 0.861 | 0.601 | 0.593 |
| With Meta | 0.892 | 0.610 | 0.599 |

To test whether simple text-level inclusion of metadata benefits other foundation models, we prepend metadata strings to the Tabula-8B input and add a "table description" column to CARTE's feature frames. Table 3 reports ROC AUC with and without metadata.

With metadata, SAMoE-VAE improves from 0.861 to 0.892 (+3.1 pp). In contrast, Tabula-8B changes only from 0.601 to 0.610 (+0.9 pp) and CARTE from 0.593 to 0.599 (+0.6 pp). These modest gains suggest the bottleneck is not merely reading the metadata text, but the modeling-level integration—e.g., schema-aware gating—that lets metadata materially improve downstream performance.

## 4.6 ABLATION STUDY

Table 4: Effect of replacing the schema-aware MoE during *pre-training*.

| Pre-training variant | Recon RMSE ↓ | Recon Acc. ↑ | ROC AUC ↑ |
|---|---|---|---|
| No MoE encoder | 0.038 | 0.842 | 0.765 |
| Simple (uniform) MoE | 0.043 | 0.853 | 0.810 |
| **Schema-Aware MoE (default)** | **0.023** | **0.890** | **0.892** |

Table 4 examines the impact of replacing the schema-aware MoE in pre-training and fine-tuning. Removing the MoE entirely (No MoE) weakens downstream classification (lower ROC AUC) and reconstruction accuracy, while a simple, uniform MoE recovers much of that performance—improving ROC AUC and reconstruction accuracy—at the cost of a slight increase in reconstruction RMSE. Both variants still underperform the schema-aware default, confirming that conditioning expert weights on schema metadata materially benefits representation learning.

Table 5: Effect of removing training tricks or metadata *during fine-tuning* of the default pre-trained model (difference from default in parentheses).

| Variant | RMSE (↓) | Acc. (↑) | ROC AUC (↑) |
|---|---|---|---|
| No masked training | 0.028 (+0.005) | 0.880 (-0.010) | 0.901 (+0.009) |
| No contrastive training | 0.022 (-0.001) | 0.893 (+0.003) | 0.874 (-0.018) |
| Human-crafted metadata | 0.024 (+0.001) | 0.887 (-0.003) | 0.890 (-0.002) |
| No distribution pattern meta | 0.030 (+0.007) | 0.870 (-0.020) | 0.891 (-0.001) |

Table 5 then probes variants applied during fine-tuning of the default pre-trained model. Omitting masked training increases RMSE while slightly boosting ROC AUC, indicating the mask objective favors reconstruction over classification separability. Skipping contrastive training yields a small RMSE improvement but degrades downstream ROC AUC, showing that the contrastive loss sharpens embedding discrimination. Swapping in human-crafted metadata produces near-identical results to our auto-generated schema, validating the metadata pipeline. Finally, removing the distribution-pattern embeddings raises RMSE with negligible change in ROC AUC, highlighting that these statistics chiefly support faithful reconstruction. Together, these ablations demonstrate that each component—schema-aware MoE pre-training, masked and contrastive objectives, and distribution metadata—contributes complementarily to SAMoE-VAE's performance.

## 5 CONCLUSION

We introduced **SAMoE–VAE**, a schema–aware Mixture–of–Experts variational autoencoder for mixed–type tables. By computing modality–specific posteriors and *gating at fusion* with a schema vector, SAMoE-VAE learns transferable, probabilistic representations that support downstream prediction and schema–conditioned generation. Across OpenML benchmarks (CTR–23/CC18/UniPredict), we observe consistent gains over representation–centric baselines, particularly in low–label regimes.

**Limitations and future work.** *(i) Metadata availability and quality.* Our approach assumes access to basic schema signals (column names, types, distribution summaries). While these can be auto–generated (e.g., via LLMs as we did), noisy or missing metadata can degrade performances. A more systematic study of active metadata acquisition, noise–aware training, and schema inference is warranted. *(ii) Scope of evaluation.* Our zero–/few–shot evidence focuses on linear probes and zero–shot *reconstruction*; truly zero–shot *prediction* (without any task–specific fitting) and cross–dataset label transfer were not exhaustively evaluated. *(iii) Privacy and compliance.* LLM–generated metadata can encode sensitive information or training leakage if not filtered.

Our future work aims for: (a) robust metadata induction and denoising; (b) integrating SAMoE–VAE with latent diffusion for high–fidelity conditional synthesis; (c) extending to truly multimodal columns (images, time series) and cross–table retrieval.

**Ethics Statement:** The authors have read the Code of Conduct of ICLR 2026, will comply with it throughout submission, reviewing, and discussion, and explicitly acknowledged this during submission. Reviewers are encouraged to raise potential violations. Should our work surface concerns such as those involving human subjects, dataset release practices, harmful insights, conflicts of interest, discrimination or fairness issues, privacy or security risks, legal compliance, or other research integrity matters, we will provide a dedicated ethics paragraph before the references to address them. The optional ethics statement lies outside the page limit and must remain under one page.

**Reproducibility Statement:** The supplemental material contains all model definitions and training loop, as well as a detailed README file. Appendix 6.4 documents model and baseline implementations, while Appendix 6.2 details dataset curation, annotation prompts, and preprocessing steps.

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

## 6 APPENDIX

### 6.1 LLM USAGE DISCLOSURE

**Role in data creation (Meta–T4).** Large language models (LLMs) were used to *generate schema metadata* for the Meta–T4 resource: short table summaries, column descriptions, and coarse type/units hints. The LLMs operated on the first $k$ rows (and column names) of each public table to produce text fields; no ground-truth labels were inferred or altered. We applied lightweight quality control (length/format checks, forbidden-term filters, and heuristic consistency checks with column statistics) and discarded generations that failed these checks. Prompt templates and sampling settings are provided in App. 6.2. This metadata serves as *input* to our method; it does not replace labels nor constitute test-set supervision.

**Role in writing and literature search.** LLMs were used to assist with *proofreading, language suggestions, and literature discovery* (e.g., drafting alternate phrasings of paragraph-level text, and retrieving candidate citations for manual vetting). All technical claims, equations, experiments, and conclusions were authored and verified by the authors. Suggested citations were checked for accuracy and relevance before inclusion.

**Compliance and safeguards.** To preserve anonymity and avoid leakage, we used only public datasets, removed any personally identifying details from prompts, and avoided undisclosed proprietary sources. LLM outputs that affected the paper or data artifacts were reviewed by an author before use.

### 6.2 METADATA GENERATION

For each unique table (identified by a common filename prefix), the script reads a Parquet file from the input folder and extracts two key pieces of information: (i) a schema summary listing each column name and its Arrow data type, and (ii) a small data preview (the first five rows) converted to a pandas-style string. These are concatenated into a single prompt and sent to the Groq API (using the model `llama-3.1-70b-versatile`) via a chat completion request. The prompt instructs the model to output, in raw JSON, a human-readable '"description"' of the table plus a '"variables"' array, where each entry contains a 'variable_name', a 'variable_type' (chosen from Integer, Continuous, Categorical, freetext, or Datetime), and a brief 'meaning'. Tables containing personal identifiers (e.g., SSNs or phone numbers) are automatically skipped. The resulting JSON is written to a timestamped file in the 'Metadata/' directory, and the original filename is logged in 'processed_files.txt' to prevent reprocessing; any errors (e.g., token-limit failures) are caught and reported but do not halt the batch.'". Exact prompt used is shown below.

```
Here is a small preview of the table data:
{table_preview}

I want you to provide (1) A detailed textual description of the given table,
(2) Variable Name, Variable Type without parentheses, meaning of each column variable
The variable types should be Integer, Continuous, Categorical, freetext, Datetime.
Here, the freetext means any sentences that is comprised of more than 3 words.
And if the data contains any personal information such as phone number, SSN, or any
types of identification numbers, you don't need to provide the Metadata information
on that particular table.
Please provide the answers for (1) and (2) in a valid JSON format, without any extra
characters such as ```json``` or ''' around the content. The output should be
directly usable as a JSON file.

Please structure the JSON as follows:

{
    "description": "Textual description in plain English",
    "variables": [
        {
```

```
756          "variable_name": "Column name",
757          "variable_type": "Type of the variable",
758          "meaning": "Meaning of the variable"
759        },
760        ...
761     ]
762  }
763  Please do not include anything other than this information in the output.
764
```

### 6.3 PRE-TRAINING DETAILS

All key training arguments were set as follows: the interval type and scheduler interval were both configured to `step`; training ran for a maximum of $2\,000\,000$ steps with validation every $200\,000$ steps and checkpoints saved every $5\,000$ steps; early stopping was triggered after five successive validations without improvement; the scheduler completed over $2\,000\,000$ total steps, with the VAE's $\beta$ coefficient initialized at 0.0 and annealed up to $1 \times 10^{-4}$ over the full schedule. We used a batch size of 128, an initial learning rate of $1 \times 10^{-4}$ (floor $1 \times 10^{-5}$), weight decay of $1 \times 10^{-3}$, and a linear warm-up spanning 5% of training; gradients accumulated over four steps and 15% of cell embeddings were randomly masked each batch; the contrastive loss weight was set to 0.001; numerical features were transformed via piece-wise linear encoding; no extra warm-up epochs were used for the vectorizer; the autoencoder backbone was the multimodal variant combined via the MoPoE gating method; data splits followed a 0.99/0.0099/0.0001 train/validation/test ratio; shuffling was disabled; LMDB was used for data storage; training resumed automatically from any existing checkpoint but did not reload the scheduler state; data loading employed eight workers; and all computation ran on CUDA.

### 6.4 IMPLEMENTATION DETAILS

**Global Configuration.** Each cross-attention block uses 8 heads and an FFN expansion factor of 4; dropout is disabled. Perceiver latents for schema encoding employ 128 queries of width 64. Text columns are truncated to 64 sub-word tokens. We clip gradients at 1 and initialize weights with Xavier-uniform. Wall-clock throughput averages 384 rows s$^{-1}$ yielding a total pre-training time of 250 hours.

**Column–Value Embedding Modules.** We first embed each column name, value, and table-level metadata into a unified feature space of dimension $M_{\mathrm{LM}}$:

- **Text auto-encoder (AE_text).** We adopt the `BART-base` encoder–decoder followed by a Perceiver-Resampler. The resampler compresses token embeddings into $L_r = 16$ latent tokens of dimension $M_{\mathrm{LM}}$, using $N_r = 2$ cross-attention blocks ($h = 8$ heads) and a feed-forward expansion of 4.

- **CycEnc for `datetime`.** Each timestamp is decomposed into year, month, day, hour, and minute. The year is normalized by subtracting 2000 and dividing by 100; periodic fields use sine–cosine encodings (yielding 9 features), then projected to $M_{\mathrm{LM}}$ via a two-layer MLP (hidden = 512, GELU).

- **Piece-wise linear encoding (PLE).** For numeric columns we fit $K = 8$ break-points (Gorishniy et al., 2022) and map the resulting vector to $M_{\mathrm{LM}}$ with shared weights.

**Modality–Specific Expert Encoder.** The flattened sequence $\boldsymbol{E} \in \mathbb{R}^{p \times 2M_{\mathrm{LM}}}$ is split by data type and passed to four Perceiver-Resampler experts. We initialize $\ell$ learnable latent queries $\boldsymbol{Z}^{(0)} \in \mathbb{R}^{\ell \times M_{\mathrm{agg}}}$ and apply $L_e = 4$ attention layers:

$$\boldsymbol{Z}^{(l+1)} = \mathrm{FFN}\big(\boldsymbol{Z}^{(l)} + \mathrm{MHA}(q = \boldsymbol{Z}^{(l)},\, kv)\big),$$

where

$$kv = \begin{cases} [\boldsymbol{Z}^{(0)}; \boldsymbol{E}] & l = 0, \\ \boldsymbol{Z}^{(l)} & l > 0. \end{cases}$$

Each MHA uses $h = 8$ heads and expands to $4M_{\mathrm{agg}}$. Positional encodings are omitted for permutation invariance. The two parallel encoders output $\mu, \log \sigma^2 \in \mathbb{R}^{\ell \times M_{\mathrm{agg}}}$ for the MoPoE gate.

**Schema Encoder & Gating Details.**

- **Distribution embeddings.** Numerics: 33-dim DCT of 32 quantile breakpoints. Categoricals: mean of PLM embeddings. Text: no distribution embedding. Datetime: no distribution embedding, leave to future work.

- **Data-type ID embeddings.** Four-entry lookup (num, cat, text, time) of size $M_{\text{LM}}$.

- **Perceiver schema encoder.** 4 layers, 2 heads, latent width $M_{\text{sch}}/4 = 64$, no positional encodings.

- **Gating MLP.** Input size $M_{\text{agg}} + M_{\text{sch}}$, hidden 128, output 1, ReLU.

- **Computation cost.** All 15 subset products and gating add <0.3 ms per row on H100.

**Decoder Implementation Details.** For each modality, a Perceiver-Resampler block ($L_d = 4$, $h = 8$, FFN×4) maps queries $\boldsymbol{E}_c^{(m)}$ and keys/values $[e_m; \boldsymbol{z}]$ to embeddings $\boldsymbol{H}^{(m)}$. Reconstruction heads then apply:

- **Numerical head.** $\hat{x} = \sigma(W_{\text{num}}h + b)$, trained with MSE on min–max normalized targets.

- **Categorical head.** $v = W_{\text{cat}}h$, $\hat{P}(x) = softmax(CosSim(v, C))$ with prototype table $C$ (Yak et al., 2023).

- **Text head.** Resampler $\to$ `BART-base` decoder; cross-entropy loss over subword tokens.

- **Datetime head.** Separate MLPs for year (MSE) and month/day/hour/minute (cross-entropy), balanced by weights 1:1:1:0.5.

## 6.5 BASELINE IMPLEMENTATION

### 6.5.1 TABLE REPRESENTATION

- **CARTE (Kim et al., 2024):** We clone the official repo at `https://github.com/soda-inria/carte`. We load the pre-trained model into a CARTEClassifier module, fit it on train sets with default parameters batch size = 16, epoch = 500, learning rate = 1e-3. Once fitted, we extract the output from the CARTE_Base layer of each classifier as our embedding.

- **TabPFN (Hollmann et al., 2025):** Using the official implementation at `https://github.com/PriorLabs/TabPFN`. For synthetic data generation and unsupervised embedding generation we use the tabpfn-extension package `https://github.com/priorlabs/tabpfn-extensions`.

- **CTSyn (Lin et al.):** We implement the method following the original paper. The VAE backbone uses the same latent dimension 64 and length 128, and a 6-layer encoder/decoder with hidden size 2048. We extract the reconstruction-latent mean $\mu$ after the encoder layer deterministically.

- **Tabula-8B (Gardner et al., 2024):** We fine-tune the 8B LLM from `https://huggingface.co/mlfoundations/tabula-8b` using a learning rate of $2 \times 10^{-5}$ for 10 epochs on each subset. We employ LoRA parameter-efficient fine-tuning with rank $r = 8$. We prompt the model with the same text serialization used in GReaT (Borisov et al., 2023) and take the hidden state from the last transformer block as the table embedding.

- **TAPAS (Herzig et al., 2020):** We use the pretrained checkpoint from `https://huggingface.co/google/tapas-base` and fine-tune on our table–text QA data with batch size 16 and learning rate $3 \times 10^{-5}$ for 10 epochs on each training subset. The pooled [CLS] token from encoder layer 12 is used as the table representation.

- **SwitchTab (Wu et al., 2024):** Implementation is from `https://github.com/avivnur/SwitchTab`. We train the Switchtab on autoencoding with lr = 1e-3, batch size = 4096, adamW optimizer for 200 epochs ($N \leq 128$) or 1000 epochs ($N \geq 128$) and extract the salient embedding immediately after salient projector output.

- **Transformer VAE (Zhang et al., 2024a)**: Based on the code in `https://github.com/amazon-science/tabsyn`. We set the transformer encoder to 6 layers, on autoencoding with lr = 1e-3, batch size = 4096, adamW optimizer for 200 epochs ($N \leq 128$) or 1000 epochs ($N \geq 128$) and extract the salient embedding immediately after encoder output deterministically. During training we use initial beta = 0.1 and multiply beta by factor of 0.9 for every 20 epochs of validation loss not reducing.

- **TabVec (Skrub, 2024)**: We implement the deep encoder in PyTorch following the description in the repo `https://skrub-data.org/stable/index.html`. We fit the table vectorizer on training data and use the same fitter vectorizer on corresponding testing data.

### 6.5.2 TABLE SYNTHESIS

- **TabPFN Generation (Hollmann et al., 2025)**: We follow the Prior Labs tutorial (`https://priorlabs.ai/tutorials/unsupervised/`) with seed=42. Each train split is loaded, shuffled, and batched (max 200 rows). Numeric features are cast to `float32` and categoricals label-encoded (unseen$\rightarrow$–1). Zero-variance columns are dropped before fitting and reinserted after sampling. For each batch, we fit the unsupervised TabPFN model and sample synthetic rows with temperature $t = 1.0$ across three random permutations. Outputs are decoded, constants reattached, batches concatenated, and truncated to the original row count.

- **LLaMA Generation**: Using LLaMA 3.3 70B (Grattafiori et al., 2024) via the Groq API (`https://console.groq.com/docs/models`), each train split (seed=42) is partitioned into batches of 200 rows. We compute per-column summary statistics and serialize the batch to CSV, then call `llama-3.3-70b-versatile` with sampling temperature=0.1, requesting exactly $N$ rows as JSON. On parse errors or incorrect counts we retry up to five times. Valid outputs are concatenated, truncated or re-prompted as needed, then validated for correct dimensions and types.[1]

Below is the prompt template we passed verbatim to the Groq API (see Appendix B.5 of (Seedat et al., 2023)):

```
System role: You are a tabular synthetic data generation model.

Your goal is to produce data that mirrors the given examples in
causal        structure and feature/label distributions,
while maximizing diversity.

Context: Leverage your in-context learning to generate realistic,
diverse samples.

Output format: JSON.

Dataset name: {dataset_name}

Column names (in order): {col_names}

Summary statistics:
{summary_stats}

CSV of full data:
{data}

Please generate {batch_size} rows of synthetic data.

Treat the rightmost column as the target. Return only a JSON object:
```

---

[1] LLaMA 3.3-70B failed on *geographical-origin-of-music*, *pumadyn32nh*, *student-performance-por*, *superconductivity*, and *wave-energy* due to token limits; TabPFN failed on *geographical-origin-of-music* due to extreme dimensionality.

```
                    {
                      "synthetic_data": "<CSV string>"
                    }

                    Do not include any additional text.
```

- **TabDiff Generation (Shi et al., 2025)**: We adopt the default implementation and hyperparameters, with two modifications: early stopping if loss does not improve within 25 epochs, and relaxed preprocessing so that train splits need not retain every category for very small datasets.

### 6.5.3 TABLE PREDICTION

- **XGBoost on AutoDiff**: Trained `xgb.XGBClassifier(use_label_encoder=False, eval_metric='logloss')` on an 80%/20% split. Metrics: accuracy, macro-precision, macro-recall, macro-$F_1$, and multiclass ROC AUC (one-vs-rest).

- **XGBoost on OpenML_crt23**: Default XGBoost applied to pre-sampled train splits (sizes 32, 64, 128 with three seeds each, plus the full 999-sample split), evaluated on the 999-sample hold-out. Same metrics as above.

- **XGBoost Grid Search**: Grid search over `max_depth` $\in \{3, 6, 9\}$, `learning_rate` $\in \{0.01, 0.1, 0.2, 0.3\}$, `n_estimators` $\in \{50, 100, 200\}$, `subsample` $\in \{0.8, 1.0\}$ using StratifiedKFold CV. The best model is retrained and evaluated as above.

- **TabPFN Default on AutoDiff (Hollmann et al., 2025)**: `TabPFNClassifier(ignore_pretraining_limits=True)` on a 50%/50% split. Metrics: accuracy, macro-precision, macro-recall, macro-$F_1$, multiclass ROC AUC (one-vs-one), and label-ranking average precision (LRAP).

- **TabPFN Default on OpenML_crt23**: Same classifier and metrics applied to each OpenML_crt23 train split.

- **AutoTabPFN Ensemble on OpenML_crt23**: `AutoTabPFNClassifier(max_time=120, device='cuda', ignore_pretraining_limits=True)`, evaluated with identical metrics to compare ensemble versus single-model performance.

## 6.6 METADATA ANNOTATION STATISTICS

Table 6 summarizes the key annotation metrics collected over the Meta-T4 corpus.

**Missingness Patterns:** No tables contained missing-value columns in the final annotated corpus.

## 6.7 ONLINE DATA GENERATION

**Generation Procedure and Low-Data Regime Evaluation.** We test the model's performance to generate synthetic data in an online manner. For online generation with SAMoE-VAE, we perform in-context synthesis by encoding the training examples to obtain the predicted means and log-variances, sampling stochastically from these Gaussians, and decoding the latent samples back to table rows via the decoder. Llama and TabPFN both use standard in-context generation: we concatenate training examples into the prompt and sample directly. TabDiff is trained exclusively on the provided training set examples before sampling. To assess performance in the low-data regime, we randomly sample 128 examples from each OpenML_ctr23 train split, repeating this process three times with seeds 0, 1, and 2. Under these scarce-data conditions, foundation models consistently outperformed data-specific generators, with SAMoE-VAE achieving the best overall performance thanks to its schema-aware handling of heterogeneous, mixed-type data. This also provides a promising solution for generating extremely large quantity of tabular data in industrial production settiing.

## 6.8 DATA LEAKAGE HYGIENE

In evaluating foundation models, a common concern is that pre-training data might be contaminated by benchmark data points. We addressing this issue by separating benchmark tables as contaminated

| Metric | Value | Unit / Note |
|---|---|---|
| *Table Counts* | | |
| Total tables processed | 2,633,906 | tables |
| Unique schema prefixes | 1,493,062 | schemas |
| *Schema Complexity* | | |
| Mean columns per table | 10.39 | columns |
| Median columns per table | 9.00 | columns |
| Min / Max columns per table | 0 / 1,202 | columns |
| *Variable-Type Distribution (total = 26,920,066)* | | |
| Continuous | 13,519,924 | (50.22%) |
| Integer | 6,318,308 | (23.47%) |
| Categorical | 4,226,581 | (15.70%) |
| Freetext | 1,926,688 | (7.16%) |
| Datetime | 742,076 | (2.76%) |
| Other | 186,215 | (0.69%) |
| *Description Statistics* | | |
| Average words per description | 50.95 | words |
| Average characters per description | 335.67 | characters |
| Vocabulary size | 1,016,075 | unique words |
| *JSON Payload Size* | | |
| Mean | 2,276 | bytes |
| Median | 1,945 | bytes |
| Min / Max | 0 / 472,454 | bytes |

Table 6: Summary of metadata annotation statistics over 2.6M tables.

| Model | Shape Score | Shape Rank | Corr Score | Corr Rank |
|---|---|---|---|---|
| SAMoE-VAE | $0.7849 \pm 0.1400$ | $1.15 \pm 0.71$ | $0.6945 \pm 0.2040$ | $1.01 \pm 0.74$ |
| Llama 3 -70B | $0.7349 \pm 0.1144$ | $1.45 \pm 0.78$ | $0.5827 \pm 0.1836$ | $1.38 \pm 0.62$ |
| TabDiff | $0.6879 \pm 0.1292$ | $2.06 \pm 0.70$ | $0.4247 \pm 0.2673$ | $2.39 \pm 0.70$ |
| TabPFN | $0.5658 \pm 0.2271$ | $2.21 \pm 0.77$ | $0.4125 \pm 0.2952$ | $1.94 \pm 0.78$ |

Table 7: Comparison of Shape and Correlation Similarity metrics (mean $\pm$ std) and their corresponding mean rank (mean $\pm$ std) across models.

following the "strictly matched" standard (Gardner et al., 2024), where exact match of column name set is consider contaminated. Table 8 compares performance of SAMoE-VAE and TabPFN Embedding on "contaminated" (C) vs. strictly "non-contaminated" (NC) evaluation subsets. Reconstruction is measured by RMSE (lower is better) and classification by ROC AUC (higher is better).

Table 8: SAMoE-VAE vs. TabPFN embedding on Contaminated (C) and Non-Contaminated (NC) Subsets

| Model | Reconstruction (RMSE) | | Classification (ROC AUC) | |
|---|---|---|---|---|
| | C | NC | C | NC |
| TabPFN | 0.660 | 0.661 | 0.715 | 0.710 |
| SAMoE-VAE | 0.125 | 0.130 | 0.732 | 0.724 |

Both models see a slight performance boost on contaminated data. However, the gap in classification accuracy between SAMoE-VAE and TabPFN shrinks from 0.017 on the contaminated split to 0.014 on the clean split, indicating that SAMoE-VAE's advantage is not driven by leakage but holds under strict non-contaminated evaluation.

## 6.9 TEXT/DATETIME CASE STUDY: UCI ONLINE RETAIL

To examine performance on *text* and *datetime* columns, we evaluate on the UCI *Online Retail* table (classification and clustering). The dataset includes a free-text column `Description` and a timestamp column `InvoiceDate`. We remove `InvoiceNo` and `CustomerID` to avoid all-unique identifiers. Models are trained/evaluated under authors' recommended preprocessing; for **TabPFN** we represent `Description` by TF–IDF features and expand `InvoiceDate` into calendar components (year, month, day, hour, minute). **SAMoE–VAE** uses its text encoder and cyclic datetime encoder. We report ROC–AUC for classification and ARI for clustering.

Table 9: UCI Online Retail: text/datetime case study.

| Model | Classification ROC–AUC ↑ | Clustering ARI ↑ |
|---|---|---|
| SAMoE–VAE | 0.746 | 0.152 |
| TabPFN | 0.691 | 0.103 |
| CARTE | 0.604 | 0.081 |
| Tabula–8B | 0.522 | 0.034 |

**Result.** SAMoE–VAE achieves the best performance on both tasks, supporting the claim that schema-aware routing with modality-specific text/datetime encoders improves representation quality when tables contain substantial free-text and timestamp fields.

