# OpenReview forum: "SAMoE-VAE: A Tabular Foundation Model with a Schema-Aware Mixture-of-Experts Variational Autoencoder"
_ICLR.cc/2026/Conference — ICLR 2026 Conference Withdrawn Submission_

### Official Review · Reviewer_Udgn · 2025-10-21

**Soundness:** 3
**Presentation:** 2
**Contribution:** 3
**Rating:** 4
**Confidence:** 4

**Summary:**

The authors propose a new meta learning algorithm (SAMoE-VAE) for tabular data, specifically designed to be schema-aware and transferable. For each modality, the model first encodes it via a modality-specific method: language model for column names, text, and categorical; periodic encoding for dates; learnable periodic for numerical columns. These embeddings are then further passed through a perceiver network to get a fixed-sized vectors of Gaussian sufficient statistics, mean and covariance matrix. These statistics are then mixed pairwise via a product of experts and finally combined with weights, where the weights are computed based on the sufficient statistics coupled with the context vector. Extensive empirical evaluation on 3 suites of tasks from OpenML confirm the utility of the learned embeddings using linear probing against SOTA tabular methods.

**Strengths:**

1. **Well-motivated problem.** The problem of learning a foundational embedding model for tabular data is timely and well-motivated. In fact, just a few days ago, I was looking for a tabular embedding model for my own project. From my survey, it seemed like TabPFNv2 was the best. I am glad to find this work and cannot wait to try it out.
2. **Extensive literature review.** The paper is well-positioned. The authors have done an extensive literature review on the subject.
3. **Engineering effort.** The authors have clearly spent some time, optimising the design of the model. It's great that they provide all the details and carefully argue why they are necessary.

**Weaknesses:**

1. **Does meta learning really help?** To me, the crucial missing part is the ablation, where SAMoE-VAE is trained on a per-dataset basis, without meta pre-training. Given how well SwitchTab performs (rank #2), I am not convinced that meta learning helps, except for the case of the text-rich datasets or extremely few-shot (<16) settings, both of which are rare (imho) in practice.
2. **Potential use-cases.** Linear probing on top of the embedding still largely underperforms the SOTA on tabular, e.g., on CC18 it falls far behind XGBoost (8 years old model), and even on text-rich UniPredict it is again behind XGBoost that does not have text processing capabilities whatsoever. Generative capabilities? To my knowledge, there is no clear-cut strategy of how and where to apply tabular generative models. Also, the authors only show the reconstruction loss and don't measure if their embeddings actually improve the downstream DDPM on tabular. In this sense, maybe evaluating TabPFN on top of the embeddings to see if they improve SOTA could help.
3. **Still needs fine-tuning.** The authors advertise the method as a meta-learning algorithm. Yet, I feel unlike TabICL, TabPFN, or Limix, this cannot be used as a true zero-shot algorithm. It still requires fine-tuning, which could be expensive.

**Questions:**

1. What is happening with F1 scores in your Table 1, for all the embedding models they are low?
2. Do you fine-tune the TabPFNv2 model before extracting the embeddings? So you extract TabPFNv2 embeddings from the encoder or decoder? Do you use the cross-validation? I am a bit surprised to see TabPFNv2 embeddings fall so much behind other methods. In my own tests, they have consistently outperformed other SSL embeddings.
3. In Table 1, is my understanding correct that you fine-tune all of the embedding models (e.g. Tabula, TabVec etc)? It seems expensive.
4. Typos: (i) line 55 runaway comma (ii) line 54 Lin et al. missing year (iii) line 51 sentence beginning with "E.g."

---

### Official Review · Reviewer_D1cR · 2025-10-30

**Soundness:** 3
**Presentation:** 3
**Contribution:** 2
**Rating:** 2
**Confidence:** 4

**Summary:**

The paper proposes an architecture for learning representations of tabular data that is expected to handle the heterogeneity of different column data types. The model integrates a variational autoencoder (VAE) with a mixture-of-experts (MoE) mechanism, where each expert specializes in a particular modality—numerical, categorical, free-form text, etc. Besides, the model leverages metadata such as table descriptions and column details, which are encoded into a learned schema vector to enrich contextual understanding. The pretraining is conducted on an augmented T4 corpus, where the authors use LLM-generated metadata to enhance the diversity of the data. The model’s performance is evaluated on some benchmarks, and is compared with several existing tabular representation models.

**Strengths:**

- The combination of a VAE and MoE to handle multiple data modalities is conceptually appealing and well-motivated. The modular expert design aligns well with the natural heterogeneity of tabular data.

- The inclusion of experts for free-form text and datetime columns addresses an important gap in many current tabular foundation models, which often focus only on numerical and categorical data. Integrating table-level and column-level metadata is a good design choice that could improve contextual understanding of tabular structures. The augmentation of the T4 corpus with LLM-generated metadata is a good way to expand pretraining data and simulate richer supervision.

**Weaknesses:**

- The evaluation omits many of the strongest-performing tabular models, including RealMLP, TabM, GBDT variants, TabICL, and TabDPT. Even if the focus is on representation learning, comparisons against strong methods—especially foundational ones like TabICL and TabDPT—are necessary to contextualize the model’s effectiveness and practical relevance.

- While the paper reports results on multiple tasks (reconstruction, classification, and clustering), the motivation for including each is not fully clear, and the overall experimental setup could better highlight the representational strengths of the proposed model.

**Questions:**

- Since SAMoE-VAE includes experts for text and datetime modalities, could the author(s) provide quantitative results on datasets that contain substantial free-form text and temporal columns? How do these experts affect performance in such cases?

- Have you studied how SAMoE-VAE performance scales with model size (e.g., number of latent dimensions, number of experts, or schema-vector width)?

- Why were models such as RealMLP, TabM, TabDPT, or TabICL omitted from the main comparison?

- Could you elaborate on how well the model transfers across datasets with disjoint schemas, beyond reconstruction metrics? For example, can embeddings trained on one domain (e.g., healthcare) transfer effectively to another (e.g., finance)?

---

### Official Review · Reviewer_Btdr · 2025-11-01

**Soundness:** 3
**Presentation:** 3
**Contribution:** 3
**Rating:** 4
**Confidence:** 4

**Summary:**

The paper presents a novel schema-aware MoE-VAE for tabular data that routes modality experts using explicitly learned schema vector. This schema-aware MoE-VAE is quite interesting approach that goes beyond the current simply gating mechanisms for MoE approaches on tabular data, and allows to generae schema-conditioned representation. The paper describes in reasonable detail how the schema elements are encoded and how the schema vector is learnt. As such, the claims of foundation model as a metadata-supervised, multi-modal model are well founded with competitive results over benchmark datasets --CTR-23, CC-11, and UniPredict. The paper also introduces metadata enriched T4 version which contains LLM-generated table/column metadata, used for pretraining the model.

**Strengths:**

S1. The paper presents a clean architecture tying schema metadata to routing, and a novel probabilistic latent space via VAE.
S2. Experiments indicate consistent gains in zero-shot reconstruction and few-label settings.
S3. Thorough ablation studies that help isolate the value of schema-aware gating -- clear novelty of the paper.

**Weaknesses:**

W1. Inconsistent details: (a) pretraining time : is it 240 (line 302) or 250 hours (line 786)? (b) pretraining dataset size: 1.2 million tables or 2.3 million tables (in table 6)? (c) do date-time get distribution patterns or not (line 814)? (d) what is the metadata LLM used?, etc. These make the paper seem to be written in a hurried manner.
W2. Missing baselines: besides XGBoost, it is crucial to include CatBoost and LightGBM models in tabular tasks. Also missing are FT-Transformer and SAINT -- both fairly recent and quite important baselines.
W3. Synthesis tasks do not have utility tests such as Train-on-Synth + Test-on-Real, and Privacy metrics are completely missing.
W4. The paper claims that it encodes "missingness patterns" -- not at all clear how this is handled. This seems to be an overclaim.
W5. A serious issue is the handling of Date/Time -- a single positional encoding (sine-cosine) is standard but quite wrong when you have separated year, month, date, day-of-week, etc. For example, year or time-since-epoch is not cyclic but rather "trend" model, even for seemingly cyclic data there is enough research to show that better encodings exist than simple sine-cosine.
W6. The paper claims it covers 100% of schema in pretraining. It is a serious limitation for claims of foundational model -- it does not indicate how well the model performs when dealing with unseen schema elements, or scaling laws (does performance keep rising as you add more distinct schemas, or saturate early?). Besides this, the LLM calls could be reduced if the schema can be sampled.
W7. Another issue that makes it hard to consider it as a "foundational model" (contrary to the claim), is the fact that test splits are per-table. That is -- backbones are trained on each table’s train split and then frozen for a linear probe. Further, the evaluation does not seem to be really over zero-shot prediction across datasets (instead linear probes and zero-shot reconstruction).

**Questions:**

Please address the questions and issues raised in the Weaknesses section above.

---

### Official Review · Reviewer_K2eY · 2025-11-02

**Soundness:** 3
**Presentation:** 3
**Contribution:** 3
**Rating:** 6
**Confidence:** 5

**Summary:**

This paper proposes a variational auto-encoder for tabular data, Schema-Aware MoE-VAE (SAMoE-VAE). SAMoE-VAE incorporates column types, domain tags, distribution patterns, and other metadata directly into the routing mechanism. Through empirical studies, the paper shows the effectiveness of  SAMoE-VAE on various tasks along with ablations to show the importance of the components.

**Strengths:**

In general, the paper is well-written and easy to follow. The concept handling different datatypes separately is crucial, and the ideas of enriching with the meta-information of table (along with enriching the llm-generated pre-trained data) can be useful for further extensions on related research. The paper appropriately addresses the limitations and future works.

**Weaknesses:**

Please refer to the questions.

**Questions:**

-	I cannot find the ‘string context’ (figure1) in the main manuscript. It would be helpful to indicate it or match it with the main text.
-	What is the reason behind choosing the variational autoencoders?
-	It would be interesting to compare with a more recent model TARTE that separates the handling of different data types.
-	What are the characteristics of the datasets? (For instance, how many columns are numerical, categorical, textual, datetime, etc.,)
-	What would be the computation time for running model (excluding the pre-training)?
- It would be interesting to see the comparison of regression performances in other datasets. I think CTR-23 might not be sufficient enough to address the different data types present in the downstream datasets.
- Possibly, datasets from CARTE/TARTE or TextTabBench can be useful to further show the effectiveness of the proposed method.

---

### Note · Authors · 2025-11-22

I have read and agree with the venue's withdrawal policy on behalf of myself and my co-authors.